# Occurrence and Effects of Antimicrobials Drugs in Aquatic Ecosystems

**Ronield Fernandez [1],\*, Nieves R. Colás-Ruiz [2] , Hernando José Bolívar-Anillo [1] , Giorgio Anfuso [3],\* and Miriam Hampel [4]**

1   Facultad de Ciencias Básicas y Biomédicas, Universidad Simón Bolívar, Carrera 59 No. 59-65, Barranquilla 080002, Colombia; hbolivar1@unisimonbolivar.edu.co
2   Facultad de Ciencias Marinas y Ambientales (CASEM), Universidad de Cádiz, 11510 Puerto Real, Spain; rocio.colas@uca.es
3   Departamento de Ciencias de la Tierra, Facultad de Ciencias del Mar y Ambientales, Universidad de Cádiz, 11510 Puerto Real, Spain
4   Instituto Universitario de Investigación Marina (INMAR), Departamento de Química Física, Universidad de Cádiz, 11510 Puerto Real, Spain; miriam.hampel@uca.es
\*   Correspondence: rfernandez17@unisimonbolivar.edu.co (R.F.); giorgio.anfuso@uca.es (G.A.)

**Abstract:** Currently, thanks to the development of sensitive analytical techniques, the presence of different emerging pollutants in aquatic ecosystems has been evidenced; however, most of them have not been submitted to any regulation so far. Among emerging contaminants, antimicrobials have received particular attention in recent decades, mainly due to the concerning development of antibiotic resistance observed in bacteria, but little is known about the toxicological and ecological impact that antimicrobials can have on aquatic ecosystems. Their high consumption in human and veterinary medicine, food-producing animals and aquaculture, as well as persistence and poor absorption have caused antimicrobials to be discharged into receiving waters, with or without prior treatment, where they have been detected at ng-mg $L^{-1}$ levels with the potential to cause effects on the various organisms living within aquatic systems. This review presents the current knowledge on the occurrence of antimicrobials in aquatic ecosystems, emphasizing their occurrence in different environmental matrixes and the effects on aquatic organisms (cyanobacteria, microalgae, invertebrates and vertebrates).

**Keywords:** aquatic ecosystems; effects of exposure; sediments; surface water; marine ecosystem

## 1. Introduction

The continuous production and consumption of synthetic substances, together with the low efficiency of conventional wastewater treatment technologies, have caused the discharge of thousands of substances into aquatic ecosystems, threatening the health of the organisms living in them. Among the synthetic substances, there is a growing interest in the so-called emerging contaminants (ECs), the use of which is, in many cases, unregulated. Furthermore, such substances, which have gone unnoticed for a long time because of the difficulties that existed in their detection and quantification, can now be easily detected and monitored thanks to the development of analytical techniques [1–3].

Within ECs, antimicrobials drugs are substances capable to kill or inhibit the growth of bacteria (antibacterials), fungi (antifungals), parasites (antiparasitics) and viruses (antivirals) [4]. Antimicrobial drugs are frequently used for the treatment of infectious diseases in human and veterinary medicine [5,6]. In addition, they are used in terrestrial animal farms and aquaculture [7].

There are currently no regulations for the consumption of antimicrobials worldwide [8]. The easy access to such substances, the increment in the human population and the rise in animal protein production, has led to an increase in their consumption [9,10].

In this context, Bortone et al. [11] reported an increase in antimicrobial consumption of 65% between 2000 and 2015. Recently, the pandemic due to COVID-19 has dramatically favored the large-scale use of a group of antimicrobials for therapeutic purposes (azithromycin, ivermectin, remdesivir, favipiravir, among others) [12]. Therefore, monitoring of antimicrobial drugs is necessary for the development of guidelines to enable their regulation.

The high consumption of antimicrobials together with their low absorption by humans and animals [13] has caused them to be continuously discharged into wastewater treatment plants (WWTP) [14] or directly into surface waters. Conventional treatments applied in WWTPs are ineffective in removing many of these antimicrobial compounds [15], allowing their entry into multiple aquatic ecosystems at concentrations in the range of ng to mg $L^{-1}$ [16,17].

The presence and effects of antimicrobial compounds have been studied extensively in terms of the phenomenon of antibacterial resistance in both aquatic and human environments [18]. However, there is currently very little knowledge of the effects of antimicrobials on aquatic organisms (cyanobacteria, microalgae, invertebrates and vertebrates) [10]. Most of the available data are reduced to acute toxicity tests in which instantaneous alarming responses such as death and histopathologies are evaluated after exposure to elevated levels (mg $L^{-1}$). However, due to the continuous exposure, environmental concentrations of antimicrobials, in the range of ng to μg $L^{-1}$, could induce more subtle negative effects on living organisms which may reduce fitness or reproductive potential and thence have knock-on ecological effects [10,19,20].

Therefore, the aim of the present review paper is to show the current knowledge on consumption and concentrations of antimicrobial drugs (antibacterial, antifungal, antiparasitic and antiviral) in different environmental matrices (influents and effluents from WWTP, surface water, sediment and seawater) and especially, to review the current knowledge on the toxic effects of antimicrobial drugs on aquatic organisms (cyanobacteria, microalgae, invertebrates and vertebrates). This review brings together data on the concentrations and effects of all groups of antimicrobials, thus allowing comparisons among them.

## 2. Use and Consumption of Antimicrobials

Antimicrobials have been widely used in human [21] and veterinary medicine [22,23], food-producing animals [24] and aquaculture [25,26]. Some examples of antimicrobials and their mechanisms of action are shown in Table 1.

**Table 1.** Summary examples, mode of action and use of antimicrobials in human and veterinary medicine, food producing animals and aquaculture.

| Mode of Action | Example | Main Use | Reference |
|---|---|---|---|
| | Antibacterial | | |
| Inhibits cell wall synthesis | Penicillin G Amoxicillin | Human and animal growth promoter. Veterinary and aquaculture | [21,25,27] |
| Inhibits protein synthesis | Gentamicin | Human, veterinary and aquaculture | [21,25,27] |
| | Oxitetracycline | Human, animal growth promoter, veterinary and aquaculture | |
| | Florfenicol | Aquaculture | |
| Inhibits synthesis of nucleic acids acting on DNA girase | Ciprofloxacin | Human | [21,25] |
| Interferes with folic acid synthesis. | Sulfamethoxazole Sulfadiazine | Human, veterinary and aquaculture Aquaculture | [21,25,27] |
| | Antiparasitic | | |
| Connection to GABA and direct activatión of chlorine channels | Ivermectin | Human and veterinary | [4,21,27,28] |

**Table 1.** *Cont.*

| Mode of Action | Example | Main Use | Reference |
|---|---|---|---|
| Inhibits DNA synthesis | Metronidazole | Human and veterinary | [4,21,27,28] |
| Inhibition of polymerization of microtubules | Albendazole | Human and veterinary | [4,21,27,28] |
| Increased permeability of parasite to calcium | Prazicuantel | Human and veterinary | [4,21,27,28] |
| Antifungals | | | |
| Inhibits ergosterol synthesis | Ketoconazole | Human and veterinary | [29,30] |
| Binds to ergosterol and alteration of cytoplasmic membrane | Amphotericin B | Human and veterinary | [29,30] |
| Alteration of microtubules | Griseofulvin | Human and veterinary | [29,30] |
| Antivirals | | | |
| Blocks the active site of neurominidase of the influenza virus | Oseltamivir | Human and veterinary | [31,32] |
| Inhibits viral DNA Polymerase | Acyclovir | Human and veterinary | [31,32] |
| Inhibit reverse transcriptase | Lamivudine Zidovudine | Human | [31] |

### 2.1. Human Medicine

In human medicine, a large number of antimicrobials are used for the prevention and treatment of infectious diseases, e.g., antibacterials are frequently used for the treatment of urinary tract infections [33], antifungals for the treatment of dermatophytosis [34] antiparasitics for the treatment of trichomoniasis [35], antivirals for the treatment of Human Immunodeficiency Virus (HIV) and, recently, for the Severe Acute Respiratory Syndrome Coronavirus 2 (SARS-CoV-2) [36]. Presently, there are no worldwide regulations on the consumption of antimicrobials in human medicine; however, campaigns are carried out in different countries to reduce their consumption. For example, since 2002, Spain has been part of the European project ESAC (European Surveillance of Antimicrobial Consumption), which compiles data on antimicrobial consumption in European Union countries [37].

In 2010, countries such as India, China and the United States were the largest consumers of antibacterials globally [10]. Klein et al. [38] analyzed the yearly global consumption of antibacterials between 2000 and 2015 and found that it increased by 65% (from 21.1 to 34.8 billion defined daily doses—DDDs), while the consumption rate increased by 39% (from 11.3 to 15.7 DDDs per 1000 inhabitants per day). Among all antibacterials studied, broad-spectrum penicillins evidenced the highest global consumption in 2015. Bruyndonckx et al. [39] collected data on antibacterial consumption in Europe between the years 1997 and 2017 where the highest consumption was evidenced in Greece (32.15 DDD per 1000 inhabitants per day), followed by Cyprus and Spain with 28.8 and 25.0 DDD per 1000 inhabitants per day, respectively. The antibacterials with the highest consumption were beta-lactams (penicillins), macrolides and lincosamides and tetracyclines [39].

The consumption or prescription of antifungal compounds in humans has been little studied and reports are based on country-specific behaviors. For example, Belgium is the country with the highest consumption of systemic antifungals in Europe. In the United Kingdom, approximately 1.5 tons of azole antifungals and 85 and 0.5 kg of griseofulvin and amphotericin B were prescribed, respectively in 2018 [29]. On the other hand, in a hospital in Valencia, Spain, as a consequence of the COVID-19 pandemic, fungal infections increased and consequently the consumption of antifungals in intensive care units increased by 75% in 2020 compared to 2019 [40].

The consumption of antivirals and antiparasitics in human medicine has not been analyzed globally and information on their consumption in continents or countries is very scarce. Adriaenssens et al. [31] and Nannou et al. [41] provided data on the consumption of antivirals in Europe in 2008 and 2018 respectively. France and Italy were the coun-

tries with the highest consumption in 2008, while Portugal and Estonia showed higher consumption in 2018 over France and Italy. In both studies, antivirals against HIV and Hepatitis B virus were the most widely used antivirals. On the other hand, the pandemic due to COVID-19 increased the consumption of antivirals such as remdesivir, oseltamivir, favipiravir, remdesivir among others and antiparasitics such as ivermectin [12,42].

### 2.2. Veterinary Medicine and Food-Producing Animals

In particular, antibacterial compounds are widely used in animal husbandry to promote growth and prevent disease [43]. Pigs, poultry and cattle represent the main sources of animal protein for human consumption [44]. Regulations on antimicrobial consumption in animals range from country to country. For example, in January 2017, the U.S. Food and Drug Administration (FDA) finished implementing Guidance for Industry #213, eliminating the use of medically important antibiotics for growth promotion. In the European Union, the implementation of new guidelines, such as Regulation (EU) 2019/6, are focused to promote the rational use of antibacterials in the veterinary field [45].

According to Kovalakova et al. [10] in 2010, a total of 63,200 tons of antibacterials were consumed for livestock worldwide. Tiseo et al. [9] estimated a global consumption in 2017 of 93,309 tons of antibacterials for livestock and projected an increase of 11.5% by 2030 to 104,079 tons. These authors also evidenced that Asia was the continent with the highest consumption of antibacterials in 2017, followed by South America, Europe, North America, Africa and Oceania, while China, Brazil and the United States were the main consuming countries worldwide with a percentage of 45%, 7.9% and 7.0%, respectively.

Antibacterials for human use such as tetracyclines and penicillins are frequently used in food producing animals [27]. For example, in 2018, they represented 78% of the sales of antibacterials for livestock in the United States [24]. Similarly, Davies et al. [46] in a study on 207 sheep farms, found that the most common antibacterials were tetracyclines (57.4%) and penicillins (23.7%). Other antibacterials in animal use are shown in Table 1.

Antiparasitics represent 23% of the worldwide costs for animal health [28]. North America represents 43% of deworming costs above Europe and Latin America. Likewise, companion animals and livestock represent 86% of deworming costs above swine and poultry [28]. The antiparasitics praziquantel and niclosamide are frequently used for treatment of pets (dogs and cats) [23] and benzimidazoles (albendazole) and macrocyclic lactones (ivermectin) for treatment of horses and cattle [27,28]. Other antiparasitics for animal use are shown in Table 1. The consumption of antivirals and antifungals in animals is not widespread and no concrete data on their consumption and future estimates are available [30,32,47]. The compounds used in animals are the same as those used in humans, for example, acyclovir is used to treat feline herpes [32], while ketoconazole is used for the treatment of mycoses in dogs [30]. Other antivirals and antifungals for animal use are shown in Table 1.

### 2.3. Aquaculture

In aquaculture, as in livestock and poultry farming, antimicrobials are used to prevent and control infections by eliminating pathogens that can affect populations [25]. It is difficult to determine the global figures of antimicrobial use in aquaculture, as it is subject to variations in each country, authorized compounds, diversity of species and their mode of farming [48]. For example, Vietnam is the country with the highest number of authorized antibacterials (30), followed by Chile (19) and South Korea (17). The antimicrobials used in aquaculture are mostly antibacterial compounds (Table 1). Within this group, oxytetracycline, florfenicol and sulfadiazine are the most commonly used [25,48].

Schar et al. [26] estimated a global consumption in 2017 of 10,259 tons and estimated an increase in consumption of 33% to 13,600 tons by 2030. China is the largest producer and consumer of antibacterials worldwide, followed by Vietnam [48]. In Latin America, Brazil and Chile are the largest consumers and in Europe is Norway [48].

## 3. Sources of Antimicrobials in Aquatic Environments

The presence of antimicrobials in aquatic ecosystems (Figure 1) is essentially linked to the excretion of unmetabolized antimicrobials by humans and animals [49], direct introduction by aquaculture [50] and inadequate disposal at the hospital level [49].

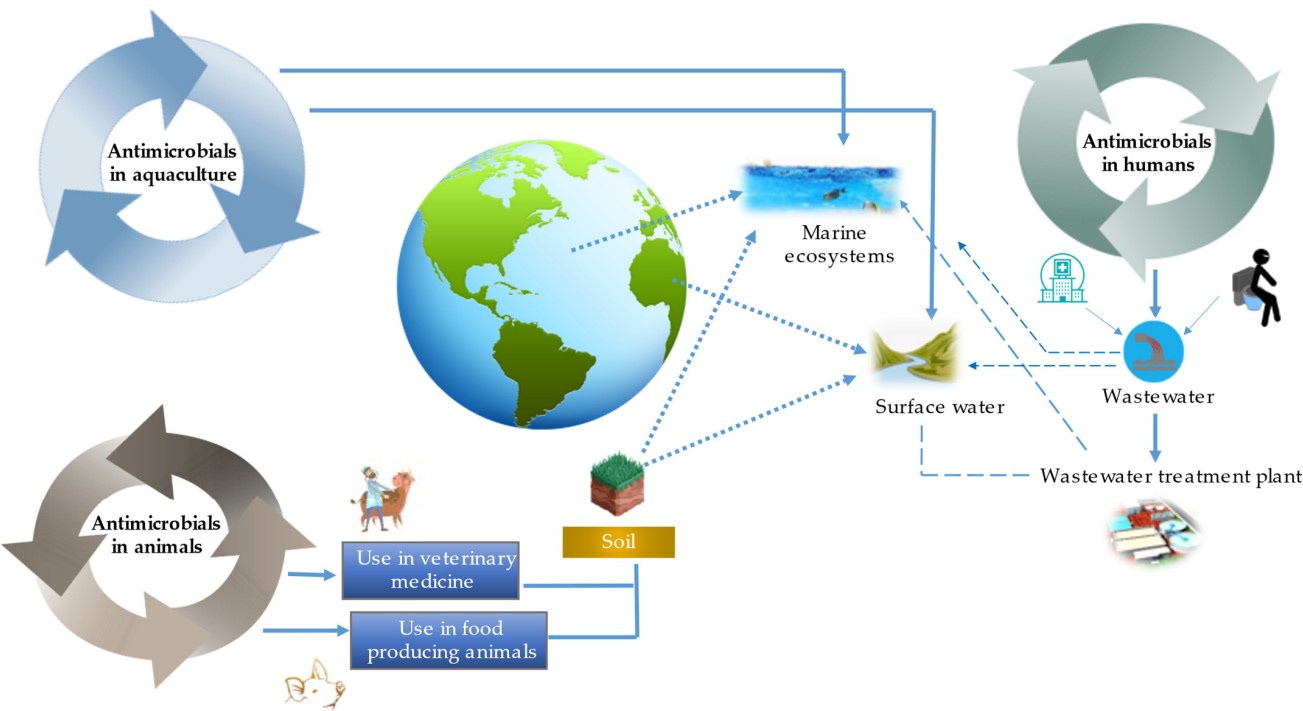

**Figure 1.** Sources of antimicrobial compounds in the environment. Human and veterinary medicine and food-producing animals such as livestock and fish are the main sources of antimicrobials in aquatic ecosystems.

Domestic and hospital wastewaters are the first destination of antimicrobials used in human medicine (Figure 1). These can be discharged directly into surface waters and marine ecosystems or after being treated in WWTPs [51]. Conventional purification processes used in WWTPs, such as activated sludge, sequencing batch reactors and anaerobic digestion process have low removal rates for antimicrobials [52].

The implementation of advanced treatment processes such as membrane bioreactors, bioelectrochemical systems and constructed wetlands showed a higher removal efficiency respect to conventional treatments of different antimicrobial compounds [15,52]. In this sense, bioelectrochemical systems have the highest removal rate (>90%), followed by membrane bioreactors (>67%) and constructed wetlands (>63%). Other current alternatives, such as the use of *Lemna minor* systems, have reported elimination percentages of 100, 96 and 86.5% of antibacterials such as cefadroxil, metronidazole and cephalexin, respectively [53,54].

Soil, like wastewater, is one of the main destinations for antimicrobials used in veterinary medicine and food animals (Figure 1). This is because a high percentage of antimicrobials used in animals are excreted unchanged and reach the soil directly [55]. In addition to their presence, antimicrobials have the particularity of persisting in the soil, contributing to their release into the biosphere and their subsequent arrival in aquatic ecosystems through lixiviation [51].

In contrast to human and veterinary use of antimicrobials and their discharge into sewage systems, current aquaculture practices are responsible for the direct introduction of antimicrobials into aquatic (marine and freshwater) ecosystems [50]. The direct and untargeted introduction of these compounds into aquaculture cages in rivers and oceans leads to dispersion in adjacent waters and thus exposure of surrounding aquatic organisms [48,50].

## 4. Antimicrobial Levels in Aquatic Ecosystems

### 4.1. Influents and Effluents from WWTPs

The concentrations of antimicrobials present in WWTP influents and effluents have been compiled from studies conducted worldwide (Table 2). The maximum concentrations of antimicrobials reported in influents are generally higher than in effluents [14,17]. Likewise, the maximum concentrations of antibacterials reported in influents and effluents were higher relative to the other antimicrobial groups [14,17].

Antibacterials are reported more frequently compared to the other antimicrobials [3,56]. Most common antibacterials include beta-lactams, tetracyclines, quinolones, macrolides, sulfonamides and diaminopyrimidines [14,57]. Groups of antiparasitics, antifungals and antivirals include benzimidazoles, azoles and antiretrovirals against HIV (Table 2).

Antibacterials for human and animal use such as sulfamethoxazole and oxytetracycline have been detected in WWTP influents at concentrations up to 59.28 mg $L^{-1}$ and 1487 ng $L^{-1}$, respectively [14,17]. Likewise, concentrations of antiparasitics, antifungals and antivirals have been detected at concentrations up to 5,000,960 and 500,000 ng $L^{-1}$, respectively [3,8].

Antibacterial concentrations of up to 14 mg $L^{-1}$ have been detected in WWTP effluents [17]. Similarly, concentrations of antiparasitics such as albendazole (129 ng $L^{-1}$), antifungals such as fluconazole (950 ng $L^{-1}$) and antivirals such as zidovudine (50,000 ng $L^{-1}$) have been reported [3,8,58].

**Table 2.** Antimicrobial concentrations in aquatic environments.

| Antimicrobial | WWTP Influent (ng $L^{-1}$) | WWTP Effluent (ng $L^{-1}$) | Surface Water (ng $L^{-1}$) | River Water (ng $L^{-1}$) | Seawater (ng $L^{-1}$) | Sediment (µg kg $^{-1}$) | Reference |
|---|---|---|---|---|---|---|---|
| | | | Antibacterial | | | | |
| Penicillin | 160 | 20 | 235 | 115 | 0.4 | - | [13,14,59] |
| Amoxicillin | 33,800 | 116,400 | 1620−4950 | 940–3190 | 5–127.8 | - | [14,60,61] |
| Oxytetracycline | 75–1487 | 2.4–24 | 230 | 51.5 | 25.1 | 652 [1] | [7,10,14,62] |
| Tetracycline | 45 * | 3.2 * | 68.90 | 31.4 | 2.4–313 | 135 [1] | [7,10,17,63] |
| Doxycycline | 24–120 | 14–49 | 9.4–25 | 1.9–68 | 103 | 7.0 [1] | [7,14,62] |
| Erythromycin | 200–300 | 30–350 | 913 | 2070 | 0.13–6.7 | 67.7 [1] | [7,10,57,64] |
| Azithromycin | 80–860 | 8–190 | 235 | 455 | 168 | - | [60,64,65] |
| Clarithromycin | 122 | 8–460 | 75–91 | 250 | 0.2–9.4 | - | [7,14,62] |
| Clindamycin | 14–37 | 18–57 | 20 | - | 4.2 | - | [7,57,62] |
| Gentamicin | 14,400–19,100 | 500 | 1400 | - | - | - | [14] |
| Amikacin | 2300 | 1000 | 1200 | - | - | - | [14] |
| Ciprofloxacin | 27 * | 14 * | 990 | 641.3 | 6.9 | 1290 [1] | [7,17,60,66] |
| Norfloxacin | 450–2200 | 0.2–628 | 3–518 | 39 | 207.5 | 5770 [1] | [7,14,62] |
| Enrofloxacin | 58 | 52 | - | 5681 | 122 | 2.34 [1] | [7,57,62] |
| Sulfamethoxazole | 59.28 * | 80,000 | 585 | 1090 | 4.4 | 0.73 [1] 0.011 [2] | [7,17,60,67] |
| Sulfadiazine | 13–26 | 10–21 | 739.20 | 580 | 8.3 | 22.0 [1] | [7,13,57] |
| Trimethoprim | 31.7–1866 | 66.6–299 | 710 | 380 | 55.8 | 9.84 [1] 0.002 [2] | [7,10,57,64,67] |
| Chloramphenicol | 13–24 | 6–21 | 91.80 | 5.8 | 8.1 | 700 * | [7,13,57,62,68,69] |

**Table 2.** *Cont.*

| Antimicrobial | WWTP Influent (ng L$^{-1}$) | WWTP Effluent (ng L$^{-1}$) | Surface Water (ng L$^{-1}$) | River Water (ng L$^{-1}$) | Seawater (ng L$^{-1}$) | Sediment (μg kg$^{-1}$) | Reference |
|---|---|---|---|---|---|---|---|
| Antiparasitic | | | | | | | |
| Metronidazole | 5–5000 | 1–70 | 1–5000 | - | - | 6–2000 [1] | [3] |
| Albendazole | 464 | 129 | - | 0.4–10.92 | 4–10 | 0.59 [1] | [58,65,70] |
| Thiabendazole | 2–80 | 10–80 | - | 1.53 | 9 | 0.01 [1] | [58,65,70] |
| Ivermectina | - | - | - | 2.54 | - | 0.013 [1] | [65] |
| Antifungal | | | | | | | |
| Clotrimazole | 16 | 0.2 | 6 | 5 | - | 2.5 [1] | [8,71,72] |
| Ketoconazole | 22 | 6.7 | 11 | 1 | - | 0.49 [1] | [8,71,72] |
| Miconazole | 16 | 7.9 | 30 | 2–30 | - | 1.25–2.06 [1] | [8,71,72] |
| Fluconazole | 960 | 950 | 133 | 109 | - | 0.057 [1] | [8,71,72] |
| Antivirals | | | | | | | |
| Oseltamivir | 59–2700 | 33–159 | 0.3–17 | 12–58 | - | - | [41] |
| Zidovudine | 9000–85,000 | 95–50,000 | 70–2500 | 1–94 | - | 118 [1] | [3,41,73] |
| Acyclovir | 1100–2500 | 12–50 | 730–1500 | 58–750 | - | - | [74] |
| Neviparine | 80–5000 | 95–3000 | 8.5–7000 | 4859 | - | 15–85 [1] | [3,41] |
| Efavirenz | 700–50,000 | 50–50,000 | 0.1–900 | 134–354 | - | 4–5 [1] | [3,41] |
| Lamivudine | 900–500,000 | 110–95,000 | 70–250,000 | 20 | - | 0.6–0.7 [1] | [3,41] |

(-): unreported; (*): mg L$^{-1}$; ([1]): river sediments; ([2]): marine sediments.

### 4.2. Surface Water, River Water, Seawater and Sediments

The concentrations of antimicrobials present in surface water, river water, seawater and sediments are presented in Table 2. Antivirals showed the highest concentrations in surface waters and rivers, while antibacterials and antiparasitics in sediments. Little information is available on antimicrobial concentrations in seawater.

Antibacterials such as oxytetracycline (230 ng L$^{-1}$), erythromycin (913 ng L$^{-1}$) and trimethoprim (710 ng L$^{-1}$) have been respectively reported in surface waters in China, France and the United States [10]. In addition, the presence of antiparasitics, antifungals and antivirals has been reported worldwide at concentrations up to 5000, 133 and 250,000 ng L$^{-1}$, respectively [3,71].

Among antimicrobials, only antibacterials (0.2–313 ng L$^{-1}$) and antiparasitics (4–10 ng L$^{-1}$) have been detected in seawater [62,63,70]. In rivers, concentrations of antibacterials (1.9–5681 ng L$^{-1}$), antiparasitics (0.4–10.92 ng L$^{-1}$), antifungals (1–109 ng L$^{-1}$) and antivirals (1–4859 ng L$^{-1}$) were reported [7,14,41,65,72].

Based on their physicochemical characteristics, antimicrobials have often been reported in river and ocean sediments (Table 2) that frequently act as temporary traps of polar con-taminants. Accumulated antimicrobials can eventually be released back into rivers and oceans [7]. In sediments from rivers, sulfadiazine, albendazole, clotrimazole and zidovudine have been detected at concentrations of 22, 0.59, 2.5 and 118 μg kg$^{-1}$ respectively [7,65,71,73], while sulfamethoxazole (0.011 μg kg$^{-1}$) and trimethoprim (0.002 μg kg$^{-1}$) have been recorded in marine sediments [67].

## 5. Toxicological Effects of Antimicrobials on Aquatic Organisms

The presence of antimicrobials in aquatic environments results in chronic low-level exposure and potential effects in different organisms (cyanobacteria, microalgae, invertebrates and vertebrates) [75]. During many years, research on the effects of exposure to

antimicrobial compounds has predominantly focused on antibacterial resistance in bacteria (Kovalakova et al. [10]). Therefore, for many compounds, comprehensive ecotoxicological data are not available [76]. Most research on the effects of antimicrobials in aquatic systems is reduced to toxicity tests on freshwater model organisms such as *Microcystis aeruginosa*, *Daphnia magna* and *Danio rerio* [77–79] and only a few studies have evaluated the effect of antimicrobials on marine organisms [80]. In addition, most studies evaluate the effect of individual antimicrobials [14,71,75,81,82] but this is not the case in natural environments, where a complex cocktail is usually observed and can produce different effects on organisms [20,83]. Furthermore, up to 90% of antimicrobials are excreted in the environment unchanged or as active metabolites [13] and also undergo natural transformation processes such as adsorption in sediments or degradation (e.g., photodegradation, hydroxylation), which can give rise to new potentially toxic compounds [84]. The effects observed in aquatic organisms exposed to antimicrobials include molecular, cellular and physiological changes (Figure 2) at exposure concentrations in the range of ng $L^{-1}$-mg $L^{-1}$.

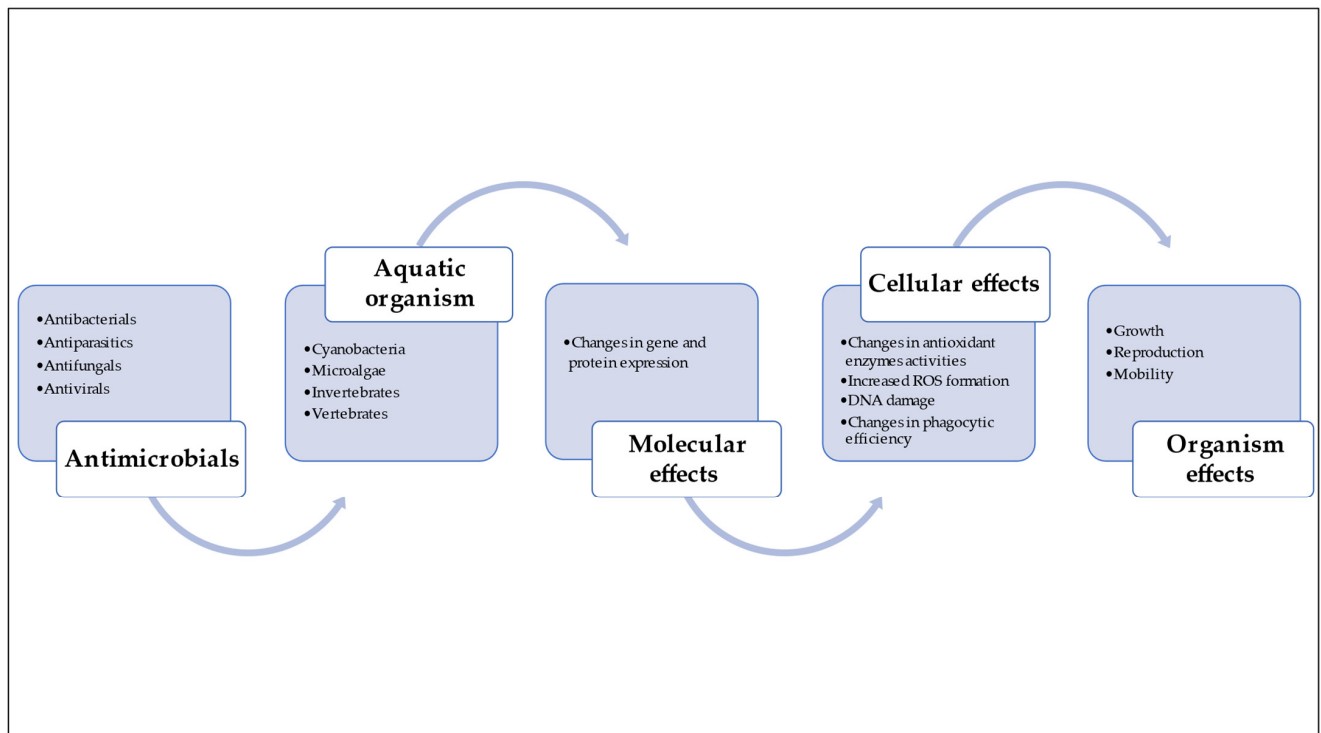

**Figure 2.** General toxicological outcome pathway in contaminant exposed organisms.

### 5.1. Fresh Water Organisms

5.1.1. Microalgae and Cyanobacteria

Within the cyanobacteria group, probably the most studied species is *Microcystis aeruginosa*. Ye et al. [85] evaluated the effect of tetracycline (TE), chlortetracycline (CTC) and oxytetracycline (OTC) on the growth of *M. aeruginosa*. The results showed mean effective concentration 10 (EC10) values at 96 h in the range of 0.63–3.02 mg $L^{-1}$ and mean effective concentration 20 (EC20) 96 h of 1.58–4.86 mg $L^{-1}$ (Table 3). Similarly, Xu et al. [86] reported mean effective concentration 50 (EC50) values of TE (2.2 mg $L^{-1}$), CTC (3.1 mg $L^{-1}$) and OTC (4.5 mg $L^{-1}$) for *Selenastrum capricornutum* and EC50 and EC10 values of OTC for *Anabaena* sp. of 2.7 and 1.5 mg $L^{-1}$, respectively.

Carusso et al. [87], by growth inhibition experiments, determined the inhibitory concentration 10 (IC10) and 50 (IC50) of CTC, OTC and enrofloxacin (ENR) on *Pseudokirchneriella subcapitata* and *Ankistrodesmus fusiformis*. The results showed a higher sensitivity of *P. subcapitata* exposed to OTC and 50% inhibitory concentrations were always lower in *P. subcapitata* compared to *A. fusiformis* (Table 3).

In addition to the evaluation of instantaneous responses such as growth inhibition in microalgae and cyanobacteria, several authors have studied the effects at concentrations similar to those reported in the environments, e.g., Liu et al. [88] analyzed the proteomic response of the cyanobacterium *Microcystis aeruginosa* exposed to amoxicillin for 30 days at a concentration of 100 ng $L^{-1}$. In total, 35 up-regulated proteins (superoxide dismutase (SOD), glutathione reductase (GR), among others) and 27 down-regulated proteins (glucose-6-phosphate isomerase, glutamine synthetase, among others) were identified. All of them are closely related to photosynthesis. Similarly, Chen et al. [89] evaluated the protein expression of *M. aeruginosa* exposed to spiramycin at 50 and 200 ng $L^{-1}$ in combination with different levels of nitrogen and phosphorus. The results showed changes in the expression of proteins related to processes such as photosynthesis, stress and cell division, such as SOD, enolase, RNA polymerase alpha and serine protease at both 50 and 200 ng $L^{-1}$ and at low and high levels of nitrogen and phosphorus.

The effect of antiparasitics such as flubendazole and fenbendazole has been evaluated in microalgae and cyanobacteria. Wagil et al. [90] reported decreased reproduction of *Scenedesmus vacuolatus* exposed to concentrations >1 mg $L^{-1}$. As for antivirals. Almeida et al. [91] and Silva et al. [92] reported growth inhibition of *Raphidocelis subcapitata* and *Microcystis novacekii* after exposure to acyclovir, efavirenz, lamivudine, zidovudine and tenofovir (Table 3). The effect of the antifungals propiconazole and tebuconazole on the growth and antioxidant response of the microalga *Chlorella pyrenoidosa* was evaluated by Nong et al. [93], identifying that doses of 100, 200 and 1000 µg $L^{-1}$ were enough to inhibit growth. These authors also observed a gradual increase of the activity of the enzyme biomarkers SOD and catalase (CAT) with concentrations ranging from 100 to 20,000 µg $L^{-1}$.

### 5.1.2. Invertebrates

*Daphnia magna* is commonly used in standard toxicity tests for crustaceans, such as the acute immobilization test (OECD 202) and reproduction test (OECD 211) [94]. Zhang et al. [95] evaluated the growth inhibition rate of *D. magna* with three antibacterials, chloramphenicol, thiamphenicol and florfenicol and their mixtures, by EC50 acute toxicity tests, at two temperatures. A significant increase in toxicity was observed with increasing exposure temperature, for example, the EC50 of chloramphenicol at 20 °C was 283.86 mg $L^{-1}$, while at 25 °C it was 85.18 mg $L^{-1}$. Similarly, the mixture of chloramphenicol and thiamphenicol showed a significant increase in toxicity at 25 °C, being the most toxic combination with an EC50 of 42.11 mg $L^{-1}$.

Luo et al. [96] evaluated the effect of lomefloxacin on the antioxidant response of *D. magna* under simulated solar radiation as a variable, observing an increase in reactive oxygen species (ROS) and decrease in oxidative stress biomarkers such as SOD (Table 3).

The effects of antibacterial compounds have also been evaluated in the bivalves *Mytilus edulis*, *Ruditapes philippinarum* and *Dreissena polymorpha* (Table 3).

The effect of antiparasitic, antifungal and antiviral drugs on *D. magna* has also been evaluated. For example, Bundschuh et al. [97] and Wagil et al. [90] evidenced alterations in motility and growth of *D. magna* due to the effect of three antiparasitics, e.g., flubendazole, fenbendazole and ivermectin, at environmentally relevant concentrations in the range of ng to µg $L^{-1}$ (Table 3) and Viera et al. [98] and Omotola et al. [81] reported concentrations of clotrimarzole (5143 mg $L^{-1}$) and lamivudine (0.1 mg $L^{-1}$) capable of causing death to *D. magna*.

**Table 3.** Main ecotoxicological effects reported for freshwater and marine species exposed to various molecules of different classes of antimicrobials at selected exposure doses.

| Antimicrobial Class | Organism | Species | Molecule | Type of Assay | Effect | Exposure Doses (mg L$^{-1}$) | Reference |
|---|---|---|---|---|---|---|---|
| Antibacterial | Freshwater cianobacteria | *Microcystis aeruginosa* | Tetracycline Chlortetracycline Oxytetracycline | Acute toxicity test, growth rate. EC10-96 h and EC20-96 h | Inhibition of growth | 0.63 1.58 1.86 4.09 3.02 4.86 | [85] |
| | Freshwater cianobacteria | *Chlorella pyrenoidesa* | Tigecycline Spiramycin Amoxicillin | Acute toxicity test, growth rate. EC50-144 h | Inhibition of growth | 6.20 4.58 >2 [1] | [82] |
| | Freshwater cianobacteria | *Anabaena cylindrica* | Tigecycline Spiramycin Amoxicillin | Acute toxicity test, growth rate. EC50-144 h | Inhibition of growth | 0.062 0.038 7.6 | [82] |
| | Fresh water microalgae | *Pseudokirchneriella subcapitata Ankistrodesmus fusiformis* | Oxytetracycline | Acute toxicity test, growth rate. IC10-96 h and IC50-96 h | Inhibition of growth | 0.07 ± 0.03 0.64 ± 0.38 0.05 ± 0.01 4.17 ± 3.79 | [87] |
| | Fresh water microalgae | *Pseudokirchneriella subcapitata* | Penicillin g Vancomycin | Acute toxicity test, growth rate. EC50-72 h | Inhibition of growth | 7114 371 | [14] |
| | Fresh water crustacean | *Daphnia magna* | Lomefloxacin | Biomarkers assay | Decreased CAT and SOD activity, induction of LPO and ROS activity. | 100 [2] | [96] |
| | | | Streptomycin Penicillin g Vancomycin | Acute immobilization test. EC50-48 h | immobilization | 487 1496 687 | [14] |
| | Marine bivalve | *Mytilus edulis* | Trimetoprim | In vitro haemocyte assay | DNA damage decreased phagocytic efficiency | 20 | [99] |
| | | | Erythromycin | Biomarkers assay | Induction of CAT and GST activity | 200 | [100] |
| | Aquatic plant | *Lemna minor* | Oxytetracycline | Biomarkers assay | Decreased plant growth | 0.001 | [101] |
| | | | Ciprofloxacin | | Increased hydrogen peroxide | ≥1.05 | [102] |
| | Bivalve | *Mytilus galloprovincialis* | Sulfamethoxazole | | Osmoregulation alteration | 0.01 | [103] |
| | Bivalve | *Dreissena polymorpha* | Trimethoprim | SCGE (single cell gel electrophoresis) assay | Genotoxicity | 5 [2] | [104] |

**Table 3.** *Cont.*

| Antimicrobial Class | Organism | Species | Molecule | Type of Assay | Effect | Exposure Doses (mg L$^{-1}$) | Reference |
|---|---|---|---|---|---|---|---|
| | Bivalve | *Ruditapes philippinarum* | Amoxicillin | Biomarkers assay | Increased CAT and decreased SOD activity | 0.4 | [105] |
| | Fresh water Vertebrate, fish | *Carassius auratus* | Erythromycin | Biomarkers assay | Decreased AChE activity, induction of SOD activity. Altered activity of anti-oxidant enzyme | 0.004–0.1 <br><br> 0.002 | [75,106] |
| | | | Roxithromycin | Biomarkers assay | Increase in AChE and SOD activity, induction of EROD activity, | 0.004–0.1 | [107] |
| | Fresh water Vertebrate, fish | *Danio rerio* | Mixture (Ciprofloxacin, ofloxacin, norfloxacin, and enrofloxacin) | | Abnormal development and histopathological changes | >37.5 | [75] |
| | | | Oxytetracycline | | Delayed hatching, inflammatory response | 0.0001–10 | [75] |
| | | | Nitrofurantoin | Biomarkers assay | Increase CAT and GST | ≥0.32 and ≥0.02 | [108] |
| | | | Sulfamethoxazole | | Increased mortality and inflammatory response | 0.260 | [75] |
| | | | Mixture (Sulfamethoxazole, sulfadiazine, sulfadimidine) | | Abnormal swimming and heart rate | 1–10 | [75] |
| | | | Clarithromycin | | Metabolic changes | 0.1 | [75] |
| | | | Erythromycin | Biomarkers assay | Significant increase of SOD, GPX and LPO activity and significant decrease of CAT | 0.01–0.1 | [109] |

**Table 3.** *Cont.*

| Antimicrobial Class | Organism | Species | Molecule | Type of Assay | Effect | Exposure Doses (mg L$^{-1}$) | Reference |
|---|---|---|---|---|---|---|---|
| | | | Oxytetracycline | Haematological parameters | Haematological alteration | 20–200 | [75] |
| | Euryhalines Vertebrates, fish | *Oncorhynchus mykiss* | Oxytetracycline | Biomarkers assay | DNA damage and altered activity of anti-oxidant enzyme | 50 | [75] |
| | | | Erythromycin | Biomarkers assay | Oxidative stress and genotoxicity | 0.0008 | [75] |
| | | | Oxytetracycline | Haematological and liver parameters | Increased ALT, AST and decreases WBC | 2500 [3] | [7] |
| | Fresh water Vertebrate, fish | *Oreochromis niloticus* | Florfenicol | Liver parameters | Decreased AST, creatinine | 5 [3] | [7] |
| | Fresh water Vertebrate, fish | *Lepomis gibbosus* | Mixture (Ciprofloxacin, ibuprofen and fluoxetine) | | Mortality | 0.01 | [75] |
| | Fresh water Vertebrate, fish | *Xiphophorus Helleri* | Norfloxacin | RT—qPCR | Genotoxicity | 0.24–6 | [75] |
| | Fresh water Vertebrate, fish | *Oryzias latipes* | Erythromycin | Biomarkers assay | Oxidative stress and genotoxicity | >1000 | [75] |
| | Fresh water Vertebrate, fish | *Cyprinus carpio* | Oxytetracycline | Haematological parameters | Increased Glu, WBC and Ht | 75 [3] | [7] |
| | Marine Vertebrate, fish | *Sparus aurata* | Oxytetracycline | Haematological parameters | Increased of WBC | 75 [3] | [7] |
| Antiparasitic | Fresh water microalgae | *Scenedesmus vacuolatus* | Flubendazole / Fenbendazole | Acute toxicity test, reproduction | Decrease in reproduction | >1 / >1 | [90] |
| | Fresh water crustacean | *Daphnia magna* | Metronidazol | Acute and chronic toxicity | Decrease in reproduction | 1000 | [110] |
| | Fresh water crustacean | *Daphnia magna* | Flubendazole / Fenbendazole | Acute toxicity test, growth rate. EC50-48 h | Inhibition of growth | 0.043–0.046 / 0.018–0.020 | [90] |

Table 3. *Cont.*

| Antimicrobial Class | Organism | Species | Molecule | Type of Assay | Effect | Exposure Doses (mg L$^{-1}$) | Reference |
|---|---|---|---|---|---|---|---|
| | Fresh water crustacean | *Daphnia magna* | Fenbendazole Flubendazole Ivermectin | Acute immobilisation test. EC50-48 h | immobilization | 0.012–0.02 0.057–0.086 0.00049–0.00072 | [97] |
| | Fresh water crustacean | *Gammarus pulex* | Fenbendazole Flubendazole Ivermectin | Acute immobilisation test. EC50-96 h | immobilization | 0.123–0.174 0.087–0.127 0.001–0.0016 | [97] |
| | Fresh water crustacean | *Asellus aquaticus* | Fenbendazole Flubendazole Ivermectin | Acute immobilisation test. EC50-96 h | immobilization | >1 >1 0.315–0.482 | [97] |
| | Fresh water Vertebrate, fish | *Danio rerio* | Doramectin | | Abnormal swimming | 0.58 | [111] |
| Antifungals | Fresh water Vertebrate, fish | *Carassius auratus* | Ketoconazol | Biomarkers assay | Increase in SOD activity and decrease in GST, EROD and AChE. | 0.002–0.02 | [71,112] |
| | | | | qPCR | Increased of *cyp1a* | 0.025–0.1 | |
| | Fresh water crustacean | *Daphnia magna* | Clotrimazole | Acute toxicity test. LC50-48 h | Mortality | 5143 | [98] |
| | Fresh water Vertebrate, fish | *Cyprinus carpio* | Clotrimazole | RT—qPCR | High *mdr1* and *mrp2* gene expression | 0.00287–0.034 | [71] |
| | | | | | Decrease of *cyp2k* and *cyp3a* | 0.00001–0.1 | |
| | Euryhalines Vertebrates, fish | *Salmo salar* | Ketoconazole | | Decrease of p-nitrophenol hydroxylase | 0.2–80 [2] | [71] |
| | | | Miconazole | | Decrease of Ethinyl estradiol | 0.1–10 | |

<div align="center">

**Table 3.** *Cont.*

</div>

| Antimicrobial Class | Organism | Species | Molecule | Type of Assay | Effect | Exposure Doses (mg L$^{-1}$) | Reference |
|---|---|---|---|---|---|---|---|
| | Fresh water Vertebrate, fish | *Danio rerio* | Clotrimazole | RT—qPCR | Increased of *fshr* and *fshβ* | 0.03–0.197 | [71] |
| | | | | RT—qPCR | Increased of *cyp17a1* and *cyp11c1* | 0.071–0.258 | |
| | | | Propiconazole | qPCR | Increased of *cyp51* and *cyp7a1* | 0.42–17.57 | |
| | | | Propiconazole | Acute toxicity test. LC50-96 h | Mortality | 12.90 | |
| | | | Difenoconazole | Acute toxicity test. LC50-96 h | Mortality | 2.34 | |
| | Euryhalines Vertebrates, fish | *Oncorhynchus mykiss* | Clotrimazole | Biomarkers assay | Increase in EROD activity | 0.0001–0.01 | [71] |
| | | | | Biomarkers assay | Decrease in EROD activity | 0.34–17.24 | |
| | | | Propiconazole | Acute toxicity test. LC50-96 h | Mortality | 5.04 | |
| | Fresh water Vertebrate, fish | *Oryzias latipes* | Fluconazole | Acute toxicity test. LC50-96 h | Mortality | >100 | [71] |
| | | *Pimephales promelas* | Ketoconazole | qPCR | Increase of *cyp11a* | 0.1, 0.3, 0.9 | [2,71] |
| | | | Propiconazole | qPCR | Increase of *cyp19*, *cyp17* and *cyp11a* | 0.005–1 | |
| Antivirals | Freshwater cianobacteria | *Microcystis novacekii* | Tenofovir | Acute toxicity test, growth rate. EC50-96 h | Inhibition of growth | 156.81–165.21 | [92] |
| | Fresh water crustacean | *Daphnia magna* | Lamivudine | Acute Mortality test 48 h | Mortality | 0.1 | [81] |
| | Fresh water crustacean | Ceriodaphnia dubia | Acyclovir Efavirenz Lamivudine Zidovudine | Acute toxicity test, growth rate. EC50-8 days | Inhibition of growth | 2529–3707 0.024–0.027 1242–1456 5370–5989 | [91] |
| | Fresh water algae | Raphidocelis subcapitata | Acyclovir Efavirenz Lamivudine Zidovudine | Acute toxicity test, growth rate. IC50-96 h | Inhibition of growth | 3249–4016 0.031–0.038 2753–3297 4969–5962 | [91] |

($^1$): g L$^{-1}$; ($^2$): µM; ($^3$): mg kg$^{-1}$; (CAT): Catalase; GST: Glutathione S-transferase; SOD: Superoxide dismutase; EROD: 7-Ethoxy thiopheneoxazolonedeethylase; AChE: Acetylcholine esterase; *mdr1*: Multi-Drug Resistance gene; *mrp2*: Multidrug resistance- associated protein 2 gene; GPX: Glutathione peroxidase; LPO: lipid peroxidation; ROS: Reactive oxygen species; (ALT): Alanina transaminasa; (AST): Aspartato transaminasa; (WBC): White blood cell; (Glu): Glucose; (Ht): Hematocrit; (*cyp1a*): Cytochrome P450 1a gene; (*cyp2k*):Cytochrome P450 2k gene; (*cyp3a*): Citocromo P450 3a gene; (*fshr*): Follicle Stimulating Hormone Receptor gene; (*fshβ*): Follicle-stimulating hormone β gene; (*cyp17a1*): Cytochrome P450 17A1; (*cyp11c1*): Cytochrome P450 11c1; (*cyp51*): Cytochrome P450 51; (*cyp7a1*): Cytochrome P450 7a1; (*cyp11a*): Cytochrome P450 11a; (*cyp19*): Cytochrome P450 19; (*cyp17*): Cytochrome P450 17.

### 5.1.3. Vertebrates

There are three main freshwater vertebrate organisms used as models in the ecotoxicological tests, all of them fish: goldfish (*Carassius auratus*), medaka (*Oryzias latipes*) and zebrafish (*Danio rerio*). Specific toxicological studies have also been conducted on other fish, such as rainbow trout (*Oncorhynchus mykiss*), flathead minnow (*Pimephales promelas*) or Indian carp (*Catla catla*).

Several antibacterials have been tested on freshwater fish, at different life stages and at decreasing concentrations from sublethal (mg $L^{-1}$) to environmentally relevant ($\mu$g $L^{-1}$). Using sublethal concentrations, Mattioli et al. [113] evaluated the risk of Nile tilapia (*Oreochromis niloticus*) exposed to florfenicol concentrations (58.73–381.8 mg $L^{-1}$) and obtained a mean lethal concentration value (LC50–96 h) of 349.94 mg $L^{-1}$. De Oliveira et al. [108] evaluated the effects of nitrofurantoin on *Danio rerio* embryos, using sublethal concentrations (0–100 mg $L^{-1}$) for the analysis of some enzymatic biomarkers. Cholinesterase, lactate dehydrogenase and glutathione S-transferase activity was induced at concentrations of 0.02 mg $L^{-1}$. Ma et al. [114] analyzed the proteomic profile of the liver of *Ctenopharyngodon idellus* fish exposed to ENR (40 mg $kg^{-1}$): they identified 3082 proteins and 103 of them were differentially abundant, 49 up-regulated and 54 down-regulated. Some of them were extremely significantly related to translation.

Using concentrations similar to environmental levels, Qiu et al. [115] studied the effects of four antibacterials, sulfamonomethoxine (SMM), cefotaxime sodium (CFT), TC and ENR at 0.01, 1, and 100 $\mu$g $L^{-1}$ on the transcriptome of *D. rerio* larvae observing that 692 (260 up-regulated and 432 down-regulated), 713 (239 up-regulated and 474 down-regulated), 592 (241 up-regulated and 351 down-regulated) and 567 (208 up-regulated and 359 down-regulated) genes were differentially expressed for SMM, CFT, TC and ENR, respectively. The genes are mainly related to steroid biosynthesis and other metabolic pathways.

Gene expression has also been evaluated in *D. rerio*, *Carassius auratus* and *Cyprinus carpio* fish after exposure to antifungals such as clotrimazole and ketoconazole [71]. Some of the differentially expressed genes were *cyp1a*, *mdr1* and *fshr* (Table 3). As for antiparasitics, a decrease in *D. rerio* swimming behavior was evidenced after exposure to doramectin (0.58 mg $L^{-1}$) [111].

### 5.2. Marine Organisms

Although antibacterials tend to bioaccumulate in marine organisms [116–118] and are nowadays a recognized threat to the marine environment [119], data on their toxicity on marine organisms are scarce, as they are usually discharged in rivers and other freshwater bodies [120].

In this context, Rodrigues et al. [121] evaluated the histopathologic effects of the antibacterials erythromycin (ERY) and OTC in the sea bream (*Sparus aurata*). *S. aurata* were exposed acutely (96 h) and chronically (28 days) to concentrations of ERY (0.0002–200 $\mu$g $L^{-1}$) and OTC (0.0004–400 $\mu$g $L^{-1}$). The results showed various alterations (circulatory, regressive, progressive and inflammatory), as well as an increase in the histopathological index of the gills of acutely exposed organisms to ERY and those chronically exposed to OTC. Similarly, Rodrigues et al. [122] evaluated the effect of ERY in *S. aurata* on some biomarker activity such as glutathione peroxidase (GPX) and glutathione reductase (GR) after exposure to concentrations of 0.3–323 $\mu$g $L^{-1}$ for 96 h and 0.7–8.8 $\mu$g $L^{-1}$ for 28 days. The results showed a decrease in GPX activity in the liver after acute exposure and an increase in the gills after chronic exposure.

Hoseini et al. [123] treated *Oncorhynchus mykiss* specimens with OTC (0 and 2.5 g $kg^{-1}$) for 2 weeks and evaluated the effects on immunological parameters, oxidative stress and enzymatic activity, recording a significant increase in serum alanine aminotransferase (ALT) and aspartate aminotransferase (AST) activities, a decrease in SOD activity and an increase in intestinal glutathione transferase (GST). Similarly, Nakano et al. [124] evaluated the effect of OTC in *Oncorhynchus kisutch* after a treatment of 100 mg $kg^{-1}$ body weight/day

orally for 2 weeks. The results showed an increase in ALT activity and total glutathione (tGSH) levels in the liver.

Other effects of antifungal compounds (azoles) on marine bivalves such as *Mytilus edulis* and fish such as rainbow trout and *Salmo salar* are presented in Table 3.

### 5.3. Toxicity of Antimicrobial Mixtures

Most research on the effects of antimicrobials in aquatic organisms has been conducted using only one compound at a time (Table 3). However, study on the effects of mixtures of antimicrobial compounds on these organisms is increasing. In this regard, Trombini et al. [20] evaluated the effect of the mixture of ciprofloxacin, flumequine and ibuprofen on the crayfish *Procambarus clarkii* at concentrations of 10 and 100 µg $L^{-1}$, obtaining alterations in immune responses and the abundance of proteins associated with biotransformation and detoxification processes in the cell (CAT and GST), as well as an increase in the expression of genes encoding antioxidant enzymes such as SOD and GPX. Jiang et al. [83] evaluated the effect of a mixture of 5 antibacterials (amoxicillin, ciprofloxacin, spiramycin, sulfamethoxazole and TE) at concentrations between 50 and 500 ng $L^{-1}$ on the biochemical, transcriptomic and proteomic responses of *Microcystis aeruginosa*. The biochemical responses showed an increase in the growth rate of *M. aeruginosa* at levels between 50–400 ng $L^{-1}$. The transcriptomic analysis revealed 206 up-regulated and 114 down-regulated genes in organisms exposed to 200 ng $L^{-1}$ and proteomic analysis identified 61 up-regulated and 25 down-regulated proteins. Differentially expressed genes and proteins were closely related to processes such as photosynthesis and carbon metabolism.

Other studies on the effects of antimicrobial compound mixtures on aquatic organisms are presented in Table 3.

### 6. Conclusions

Studies on the presence of antimicrobials in aquatic ecosystems and their effects on aquatic organisms have focused mainly on antibacterials; however, the effects of antiparasitic, antifungal and antiviral compounds in these ecosystems need to be further studied and determined.

The types of antimicrobials and the levels detected are related to the low efficiency of their removal in WWTPs, mainly in developing countries. Trade and endemic diseases also play an important role, for example, antivirals and antiparasitics are rarely detected in Europe, however, in Africa they have been detected at concentrations very close to mg $L^{-1}$, which could be related to a higher consumption of these antimicrobials in malaria and AIDS endemic countries in Africa. As for trade, it is difficult to establish a relationship between the consumption of antimicrobials and their presence in aquatic ecosystems; however, the highest concentrations detected are reported in countries with high consumption, such as China, India and the United States. In South America, there are few studies that provide information on the presence of antimicrobials in different aquatic ecosystems.

On the other hand, most of the effects are usually measured at concentrations that are not relevant from an environmental point of view (mg $L^{-1}$), and do not reflect the real behavior in aquatic scenarios. The use of molecular tools and chronic exposure tests at concentrations similar to environmental levels (ng-µg $L^{-1}$) need to be performed more frequently.

Undoubtedly, the continuous introduction of antimicrobial compounds into aquatic ecosystems is a global problem that paints a bleak picture for the future. There is an imminent need for different countries to establish standards that allow greater control over the consumption of antimicrobials (e.g., human and veterinary medicine and food-producing animals), and to implement new technologies in WWTPs and/or new wastewater treatment systems to eliminate these compounds, thus preventing their entry into aquatic ecosystems.

**Author Contributions:** Conceptualization, R.F. and M.H.; methodology, R.F., H.J.B.-A. and G.A., writing—original draft preparation, R.F., H.J.B.-A., N.R.C.-R., M.H. and G.A., writing—review and editing, G.A., M.H. and N.R.C.-R. All authors have read and agreed to the published version of the manuscript.

**Funding:** This study was funded by the Spanish Ramón y Cajal funding scheme supporting MH (contract reference RYC-2012-12217) and the Spanish Ministry of Economy, Industry and Competitiveness (MINECO) within the framework of the project: Integration of Omics Tools for the Environmental Risk Assessment of Emerging Pollutants in Marine Species of Commercial Interest, HORACIO (CTM2015-70731-R).

**Institutional Review Board Statement:** Not applicable.

**Informed Consent Statement:** Not applicable. The study did not involve humans or animals.

**Data Availability Statement:** Data supporting reported results can be found asking directly of the first author.

**Acknowledgments:** This work was supported by a fellowship of the Latin American Association of Postgraduates (AUIP) in agreement with the Simón Bolívar University of Colombia, the PROPLAYAS Network, the Caribbean Marine and Limnological Research Center CICMAR and the University of Cadiz in Spain. This work is a contribution to the PAI Research Group RNM-328 (Junta de Andalucía, Spain).

**Conflicts of Interest:** The authors declare no conflict of interest.

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
