# Peer review of "Occurrence and Effects of Antimicrobials Drugs in Aquatic Ecosystems"

_sustainability, doi:10.3390/su132313428_

Round 1

Reviewer 1 Report

The Review presented in this manuscript is well organized, scientifically sound and based on relevant and recent literature, providing a useful contribution to an emerging challenge.

Author Response

Thank you very much.

Reviewer 2 Report

In the current review article the authors are focusing on the toxicological and ecological impact of antimicrobials on aquatic ecosystems. Introduction, although brief, summarizes the key aspects of the presence and effects of antimicrobial compounds as in insufficiently studied on aquatic organisms.

2.1. Human medicine; 2.2. Veterinary medicine and food-producing animals

The new regulations regarding consumptions of such should be stated

5.1.2. Invertebrates

More examples should be stated, D. magna is merely one

A chapter of bioremediation should be made for example duckweed usage [1]

Ref
Evangelia I. Iatrou, Georgia Gatidou, Dimitrios Damalas, Nikolaos S. Thomaidis, Athanasios S. Stasinakis, Fate of antimicrobials in duckweed Lemna minor wastewater treatment systems, Journal of Hazardous Materials,Volume 330,2017,Pages 116-126,ISSN 0304-3894, https://doi.org/10.1016/j.jhazmat.2017.02.005.

Author Response

Dear Reviewer,

after an exhaustive review, it is possible to state that there are not regulations concerning the consumption of antimicrobials in human medicine. Despite of this, at some countries, campaigns are carried out to reduce their consumption.

Many countries do not show any regulation on the consumption of antimicrobials in animals and, in other countries, such regulations range a lot from country to country - examples from the United States and the European Union have been presented.

The use of Lemna minor has been added as an alternative for the elimination of antimicrobial compounds, indicating some elimination percentages.

It was added that the effects of antibacterial compounds have also been evaluated on other invertebrate organisms such as Mytilus edulis, Ruditapes philippinarum and Dreissena polymorpha).